# Firefly-inspired bipolar information indication system actuated by white light

Hanwen Huang [1,5], Jiamiao Yin[2,5], Qianwen Zhou[3], Huateng Li[1], Junying Yang[1], Yaoben Wang[1], Ming Xu [2,4] ✉ & Changchun Wang [1] ✉

The indication of information in materials is widely used in our daily life, and optical encoding materials are ideal for information loading due to their easily readable nature and adjustable optical properties. However, most of them could only indicate one type of information, either changing or unchanging due to the mutual interference. Inspired by firefly, we present a non-interfering bipolar information indication system capable of indicating both changing and unchanging information. A photochemical afterglow material is incorporated into the photonic crystal matrix through a high-throughput technique called shear-induced ordering technique, which can efficiently produce large-area photonic crystal films. The indication of changing and unchanging information is enabled by two different utilizations of white light by the afterglow material and photonic crystals, respectively, which overcome the limitations of mutual interference. As a proof of concept, this system is used to indicate the changing photodegradation level of mecobalamin (a photosensitive medicine) and unchanging intrinsic drug information with anti-counterfeiting functionality, which is a promising alternative to instantly ascertain the efficacy of medicine at home where conventional assays are impractical.

Storing unchanging information that does not change over time, even for thousands of years, has long been a significant goal for humanity. To do so, ancient Babylonians carved the Code of Hammurabi on a rock, preserving their code to this day[1]. Apart from unchanging information, changing information that changes over time, such as the age of the rock that the code was carved on, is of equal importance[2,3]. Optical encoding is easy-to-read and adjustable, making it favorable for information storage and indication[4–7]. Structurally colored materials, like photonic crystal, is a popular optical encoding material, whose color arises from the underlying periodic nanostructure and could reflect light of certain wavelength[8,9]. It is an ideal candidate for optical encoding because of its excellent stability and resistance to photobleaching. Also, the underlying sophisticated nanostructure of photonic crystals is hard to be duplicated and endows an anti-

counterfeiting feature. Therefore, structurally colored material is widely used to indicate unchanging information given its high color stability which does not fade like dyes and pigments[10,11].

However, it is still challenging to indicate changing information with structurally colored material. Incorporating luminescent material into photonic crystals is a promising approach to increase the volume of information indication. Whereas, most luminescent material can only exhibit stable light intensity based on a photophysical process, which can only indicate unchanging information[12–15]. To indicate changing information, photochromic material, which undergoes isomerization when exposed to light of a specific wavelength, could be a solution. When exposed to UV irradiation, the color change is correlated with the UV irradiation dose, which can be used to track changing information[16]. However, the UV light would interfere with the two

[1]State Key Laboratory of Molecular Engineering of Polymers, Department of Macromolecular Science, Laboratory of Advanced Materials, Fudan University, Shanghai 200433, China. [2]State Key Laboratory of Molecular Engineering of Polymers, Department of Chemistry, Fudan University, Shanghai 200433, China. [3]Institute of Microscale Optoelectronics, Shenzhen University, Shenzhen 518060, China. [4]School of Chemistry and Chemical Engineering, Shanghai Jiao Tong University, Shanghai 200240, China. [5]These authors contributed equally: Hanwen Huang, Jiamiao Yin. ✉e-mail: xu-ming@sjtu.edu.cn; ccwang@fudan.edu.cn

optical states of the material, and hence not suitable for indicating unchanging information. All in all, the above optical encoding materials can only indicate one type of information, either unchanging or changing due to mutual interference.

Fortunately, we can learn from the nature. Firefly lantern exhibits unchanging gray reflection color in the day and emission color with changing light intensities at night. This unique feature is enabled by the underlying structure of the firefly lantern, with a photogenic layer coupled with a reflection layer. The bioluminescence reaction in the photogenic layer is responsible for the changing emission intensities at night[17,18]. The reflection layer is composed of monodispersed spherical granules, which is a stable physical structure that outputs constant reflection color[19]. Inspired by this unique structure, we designed a photochemical reaction-based afterglow material that features varying light intensities. This photochemical afterglow material consists of three components: a photosensitizer, a photoenergy cache unit and an emitter. When the photosensitizer is irradiated by white light, it produces singlet oxygen ($^1O_2$). The photoenergy cache unit then reacts with $^1O_2$ and forms an unstable intermediate, which decomposes and transfers energy to the emitter, resulting in the persistent luminescence[20–22]. The photoenergy cache unit itself also serves as a consumption unit (CU) due to the unidirectional reaction with $^1O_2$ and would be consumed during white light irradiation, causing a gradual decline in the afterglow intensity. Unlike traditional afterglow material that exhibits stable afterglow intensity due to a photophysical process[23], ours relys on a photochemical process that features decreased afterglow intensity. Thus, the afterglow intensity could be further exploited to reveal the degree of light exposure, which is capable of tracking changing information.

Herein, we incorporated the photochemical afterglow material into colloidal particle slury and assembled them into photonic crystal (PC) film through SIOT (shear-induced ordering technique). Under white light, reflection color (selective reflection of white light) is displayed, which indicates unchanging information owingto the high stability of the underlying nanostructure. When white light is off, afterglow is displayed, the intensity of which is closely related to the previous degree of white light exposure and can encode changing information. The designed bipolar system is actuated by white light, as white light is turned on and off, it flips between two modes. As a proof of concept, a smart label that could indicate the light exposure history of photosensitive drugs was designed to ascertain medication efficacy, since excessive light exposure can render photosensitive medication ineffective[24]. Our smart label could indicate the unchanging information of the medicine through reflection color with anti-counterfeiting functionality while also indicate the medicine's degree of light exposure. We also present a tailorable mass production platform to indicate changing and unchanging information, which is a promising alternative to instantly ascertain the efficacy of medicine at home where conventional assays are impractical.

## Results and discussion
### Overview of firefly-inspired bipolar information indicating system
Male firefly lantern shows unchanging gray reflection color in daylight and changing emission color at night[18]. Such a unique visual effect is enabled by the underlying lantern structure (Fig. 1a). Twolayers are responsible for this distinct bipolar visual effect, i.e., the photogenic layer and the reflection layer. During the daytime, the firefly lantern displays an unchanging gray color. At night, it shows emission color with changing light intensities as a result of the bioluminescence reaction within the photogenic layer. This reaction is fueled by ATP and as the reaction proceeds, the light intensity gradually decreases with the continuous consumption of ATP.

Inspired by this particular structure, we designed a bipolar information indication system by assembling stable and unstable units into ordered photonic structure through a powerful and universal approach called SIOT. The entire process was very swift, capable of preparing photonic crystal structure in <10 s. The stable units are composed of core-interlayer-shell (CIS) colloidal particles, which are elaborately-designed mono-dispersed particles that can form periodic photonic structure and output stable reflection color. The CIS particles were fabricated via stepwise emulsion polymerization (Supplementary Fig. 1), which was detailed in the Experimental Section. The transmission electron microscopy (TEM) images (Supplementary Fig. 2) and dynamic light scattering (DLS) data (Supplementary Table 1) both suggested the growth of the particles after each step. Meanwhile, the monodispersity of the CIS particles was also verified, with the polydispersity index (PDI) of the CIS particles below 0.05, which is the prerequisite for forming a photonic structure.

The bipolar photonic crystal film was fabricated according to Fig. 1b. The unstable units consist of photosensitizer, CU and emitter, which together can output changing emission intensity through a photochemical reaction. Subsequently, a mediating molecule called propylene carbonate was designed for its high compatibility with the stable units and unstable units. The stable and unstable units were then blended with the mediating molecule and formed a particle slurry with disordered arrangement of particles. Then, a universal platform called SIOT was applied to force the particles into a periodic arrangement (Supplementary Fig. 3). The prepared photonic crystal film features a bipolar information indication system. It is worth mentioning that SIOT was a universal approach to combine stable units with unstable ones, as long as these two units are compatible with the mediating molecule. The mechanism of this system lies in the physical and chemical utilization of white light (Fig. 1c, left). When white light is on, the system is dominated by the physical utilization of white light. The periodic physical nanostructure of the photonic crystal film selectively reflects the white light and outputs unchanging reflection color due to its underlying stable units (CIS particles), which can be used to indicate unchanging information. When white light is off, the system features chemical utilization of white light, which outputs changing afterglow intensity through a photochemical reaction. The photosensitizer was previously irradiated by white light and produced $^1O_2$. CU then reacted with $^1O_2$ and formed an unstable intermediate, which gradually transferred the energy to the emitter, resulting in the afterglow emission in darkness. Meanwhile, CU would be consumed during white light irradiation, causing a changing afterglow emission intensity, which can be used to indicate changing information that tracks the degree of light exposure. As a result, the bipolar photonic crystal film can indicate both unchanging and changing information when white light is on and off, respectively (Fig. 1c, right).

### Structural and optical properties of the bipolar information indication system
SIOT was applied to assemble disordered bipolar particle slurry containing stable units and unstable units into ordered bipolar photonic crystal film. 2D USAXS (ultra-small angle X-ray scattering) pattern of the bipolar photonic crystal film (Fig. 2a, right) displayed clear diffraction spots on the outer scattering ring, which were not present in bipolar particle slurry (Fig. 2a, left). The 1D USAXS curves of bipolar photonic crystal film (Supplementary Fig. 4) showed a characteristic peak ratio $q_2/q_1 = \sqrt{3}$ and $q_3/q_1 = \sqrt{4}$, indicating a typical hexagonal close-packing on each layer[25].

The prepared bipolar photonic crystal film features high color tunability with two optical states when white light is on or off, which can be independently programmed by adjusting the compositions of stable units and unstable units. To obtain different reflection colors under white light, simply changing the CIS particle (stable unit) size would suffice. Similarly, different emission colors when white light is off could be achieved by choosing different emitters in the unstable units. For Fig. 2b(i), CIS particles that reflect red color were blended

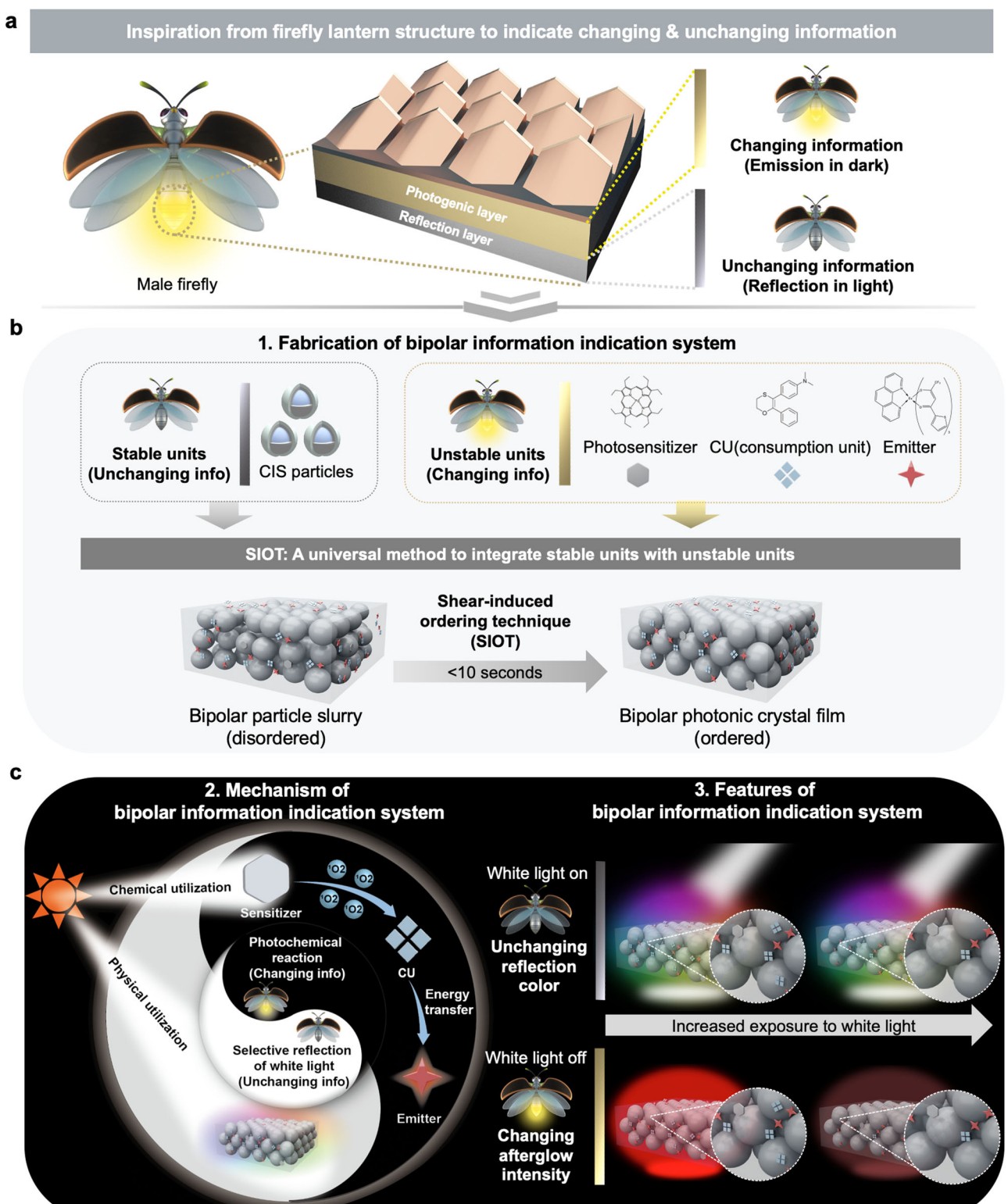

**Fig. 1 | Overview of bipolar information indication system. a** Inspiration from firefly lantern structure to indicate changing and unchanging information. **b** Fabrication of bipolar information indication system with stable and unstable units through SIOT (shear-induced ordering technique). CIS particle refers to core- interlayer-shell particle, the core is represented in blue color, interlayer dark gray and shell light gray. **c** Mechanism and features of white light-driven bipolar infor- mation indication system that can indicate changing and unchanging information.

with europium complex, CU and photosensitizer. Therefore, under white light, the system displays a stable red reflection color while white light off changing red afterglow emission color. For Fig. 2b(ii), CIS particles that reflect green color were blended with boron

dipyrromethene (BODIPY), CU and photosensitizer, and for Fig. 2b(iii) CIS particles that reflect blue color were blended with perylene, CU and photosensitizer. We can conclude that the stable units (CIS par- ticles) dictate the reflection color under white light while unstable

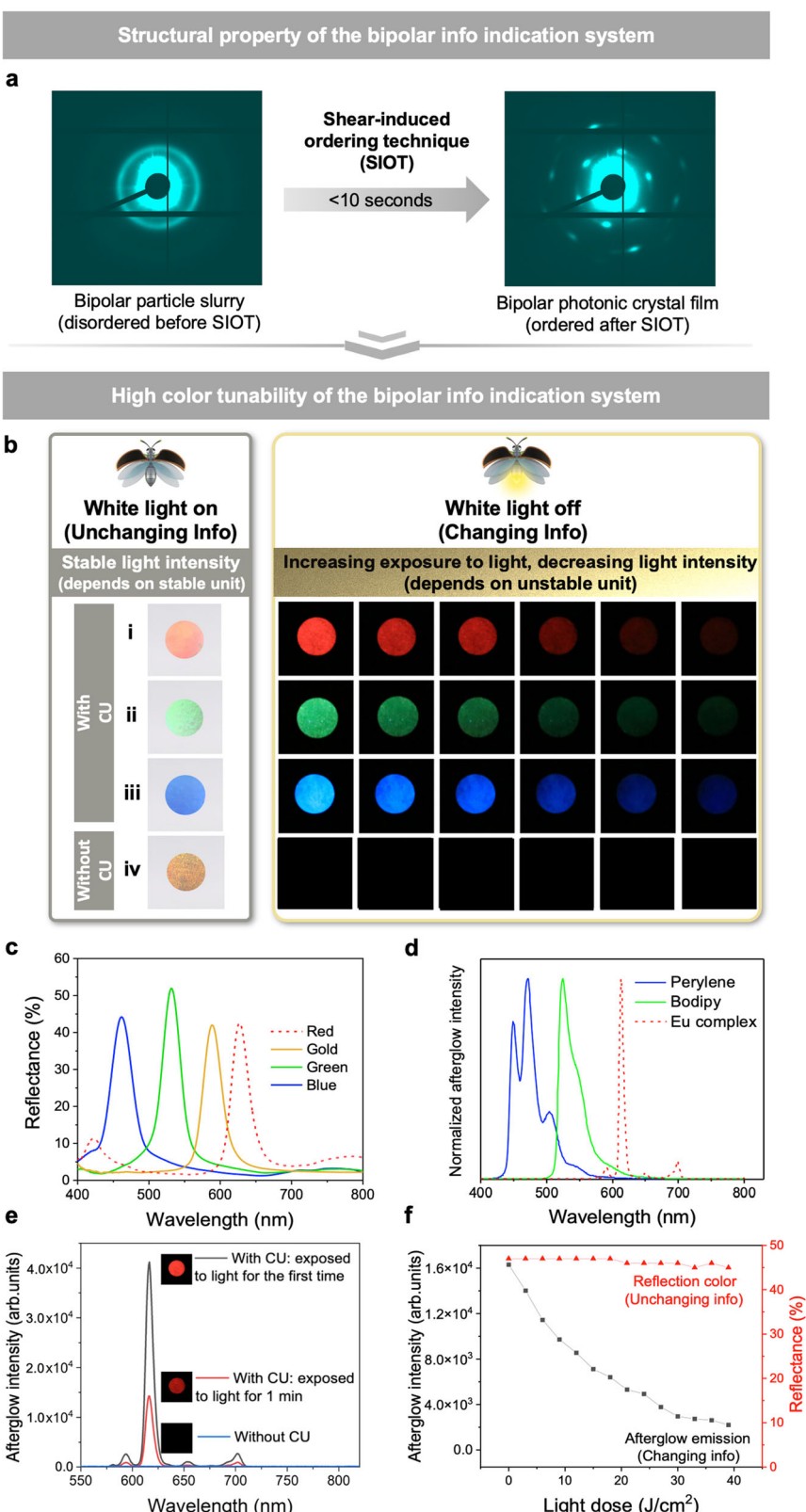

**Fig. 2 | Structural and optical properties of bipolar information indication system. a** 2D USAXS patterns of bipolar particle slurry (before SIOT) and bipolar photonic crystal (PC) film (after SIOT). **b** Digital pictures of bipolar PC films fabricated with stable units (CIS particles of different sizes) and unstable units (different compositions of emitters, photosensitizer and CU). The detailed composition for (i) is CIS particles that reflect red color blended with europium complex, CU and photosensitizer; (ii) CIS particles that reflect green color blended with boron dipyrromethene (BODIPY), CU and photosensitizer; (iii) CIS particles that reflect blue color blended with perylene, CU and photosensitizer; (iv) CIS particles that reflect gold color blended with europium complex, photosensitizer but without CU. **c** Reflectance of bipolar PC films fabricated with CIS particles that reflect different colors (under white light). **d** Normalized luminescence emission spectra of bipolar PC films fabricated with different emitters (white light off). **e** The afterglow intensity of unstable units with CU before and after irradiated with white light for 1 min. And the afterglow intensity of unstable units without CU. The emitters being europium complex. **f** The intensity of reflection color and afterglow of the bipolar PC film after being exposed to white light. The emitters being europium complex.

units determine the afterglow emission color when white light is off. The reflectance in Fig. 2c shows the different reflection colors under white light realized with different sizes of CIS particles. And the afterglow spectra in Fig. 2d show that fine-tuned afterglow emission color could be realized with different emitters in unstable units when white light is off. Their corresponding CIE diagrams were also recorded (Supplementary Fig. 5). The emission peaks and FWHW (full width at half maximum) of different emitters were also documented in Supplementary Table 2. Therefore, different combinations of reflection color and afterglow emission color in one system can be achieved simply by integrating different stable and unstable units.

In addition, the brightness of the afterglow emission gradually diminished as the system was exposed to increased level of white light (Fig. 2bi–iii). It could also be confirmed from the afterglow spectra that the afterglow intensity significantly dropped after the system was exposed to white light for 1 min (Fig. 2e). When CU is absent in the unstable units, no afterglow can be observed (Fig. 2biv and Fig. 2e). Therefore, we believe that CU plays an important role in producing afterglow and the reduced afterglow intensity is likely the result of the depletion of CU in the photochemical reaction during white light exposure. The detailed mechanism for this photochemical reaction will be explored in the next section. We then measured the intensity of the afterglow and reflection color of the bipolar PC film after being exposed to white light. The afterglow intensity decreased with the increase of light dose, while the reflection color intensity was stable and not affected by the light exposure due to the stable physical structure that does not photobleach (Fig. 2f). In addition, the reflection color was also stable under different temperatures (0 °C - 50 °C) for 1 month (Supplementary Fig. 6). As a result, the bipolar information indication system can display unchanging reflection color under white light and changing intensities of afterglow emission when white light is off.

## Mechanism of the bipolar information indication system

To understand the mechanism of the photochemical reaction within the bipolar information indication system, we break down the unstable units into three key components: photosensitizer, CU and emitter (Fig. 3a). We propose the following mechanism based on our experimental results. When the photosensitizer is exposed under white light irradiation, it absorbs photoenergy and transfers the energy to oxygen, forming $^1O_2$; CU, a $^1O_2$-reactive molecule, would then react with $^1O_2$ and produce an unstable high-energy intermediate (1,2-dioxetane) which decomposes into CU′ and transfers the energy to the emitter gradually. The absorption spectra of the photosensitizer and different emitters were documented in Supplementary Fig. 7 and S8, suggesting a good absorption among the visible range. To better understand the role of CU in the photochemical reaction, we performed a control experiment to prove the necessity of CU in the photochemical reaction. In the photochemical systems composed of different emitters but without CU, no afterglow emission can be observed. Afterglow can only be observed in the systems with CU (Supplementary Fig. 9). In addition, when we remove the emitter from the photochemical reaction, with only photosensitizer and CU present in the system, a weak afterglow emission near 380 nm was observed originated from CU′ (decomposition product of CU) (Supplementary Fig. 10). $^{13}C$ NMR spectra further confirmed the formation of CU′, with two characteristic peaks appearing around 170 ppm and 190 ppm attributed to the carbonyl of CU′ (Supplementary Fig. 11), which was also verified with the disappearance of peaks around 110 ppm and 145 ppm. In addition, before introducing CU into the system, the emitters' luminescence lifetime was between nanosecond and millisecond scale (Supplementary Fig. 12). After introducing CU, the emitters' luminescence lifetime increased to time scale of seconds, meaning it is not the direct emission of the emitters, but afterglow originated from the gradual transfer of energy from CU to the emitter (Supplementary Fig. 13). All the above

evidence suggests the afterglow is a result of consumption of CU and energy transfer to the emitter.

It can be concluded from the above results that CU plays an important role in producing afterglow. Thus, we move on to the study the effect of CU concentration on the afterglow property. To record the afterglow, we used a white light flashlight with a fixed power density of 50 mW/cm². Samples of different CU concentrations were first exposed to white light for 1 s and the light source was immediately removed. The samples' afterglow was then imaged with EMCCD (electron-multiplying charge-coupled device) as shown in Fig. 3b. After studying the afterglow imaging, we discovered two patterns: First, when the CU concentration was fixed, the afterglow intensity decreased with the increased white light exposure until it could no longer be observed; Second, when the white light exposure was fixed, samples with higher CU concentration would have a higher afterglow intensity and can take on more white light exposure until its afterglow can no longer be observed (Fig. 3c). These findings were intuitional, as CU serves as an energy donor to power the afterglow and is gradually consumed during white light irradiation, leading to a reduction of energy that can be transferred to the emitter and thus a reduced afterglow intensity. The afterglow intensities of samples with different CU concentrations under different white light doses were recorded in Fig. 3d, which further corroborated the afterglow imaging results in Fig. 3c. Finally, we investigated the stability of the bipolar information indication system in darkness. When the sample with a CU concentration of 12 mM was placed in darkness, its afterglow intensity remained nearly unchanged for 7 days. The sample was further placed in darkness for 3 months, and the remaining afterglow intensity did not experience a significant decrease, maintaining a substantial portion of its original intensity (Supplementary Fig. 14). This result proved that our bipolar information indication system has good stability in darkness, which can be further exploited for more applications.

## Applications of bipolar information indication system as a quality control label for photosensitive medicines

In clinical practice, photosensitive medicine is likely to undergo photodegradation due to negligence or a damaged package. When exposed to excessive light exposure, photosensitive medicine undergoes photodegradation, turning from effective to ineffective medicine (Fig. 4a). Mecobalamin is a typical photosensitive medicine, which will gradually transfer into hydroxy cobalamin that has no medical effect when exposed to excessive white light[26]. The chemical formula of this transition was shown in Supplementary Fig. 15. We then used HPLC (high performance liquid chromatograph) to determine the mecobalamin content under different white light doses, which was recorded in Fig. 4b. The relative content of mecobalamin was calculated from the peak area of the HPLC chromatogram of mecobalamin under different white light dose (Supplementary Fig. 16). The peak position of mecobalamin was around 6.3 min and hydroxy cobalamin 3.4 min. We can clearly observe the peak area of mecobalamin decrease and hydroxy cobalamin increase when continuously exposed to white light, proving the photodegradation of mecobalamin into hydroxy cobalamin.

After knowing the photodegradation behavior of mecobalamin, we designed a tailorable mass production platform for quality control label that can indicate the photodegradation of mecobalamin (Fig. 4c). The quality control labels for mecobalamin was fabricated through the following procedures: Unstable units and stable units were first blended and the CU concentration was tuned according to the photodegradation behavior of mecobalamin to indicate different degrees of white light exposure. The bipolar slurry was obtained after tailorable blending of stable units with unstable ones. The slurry was then continuously sheared on the ordering machine, compelling the CIS particles (stable units) to arrange into an ordered photonic crystal structure. This process was called SIOT, capable of scaling up and producing large-sized bipolar photonic crystal (PC) film.

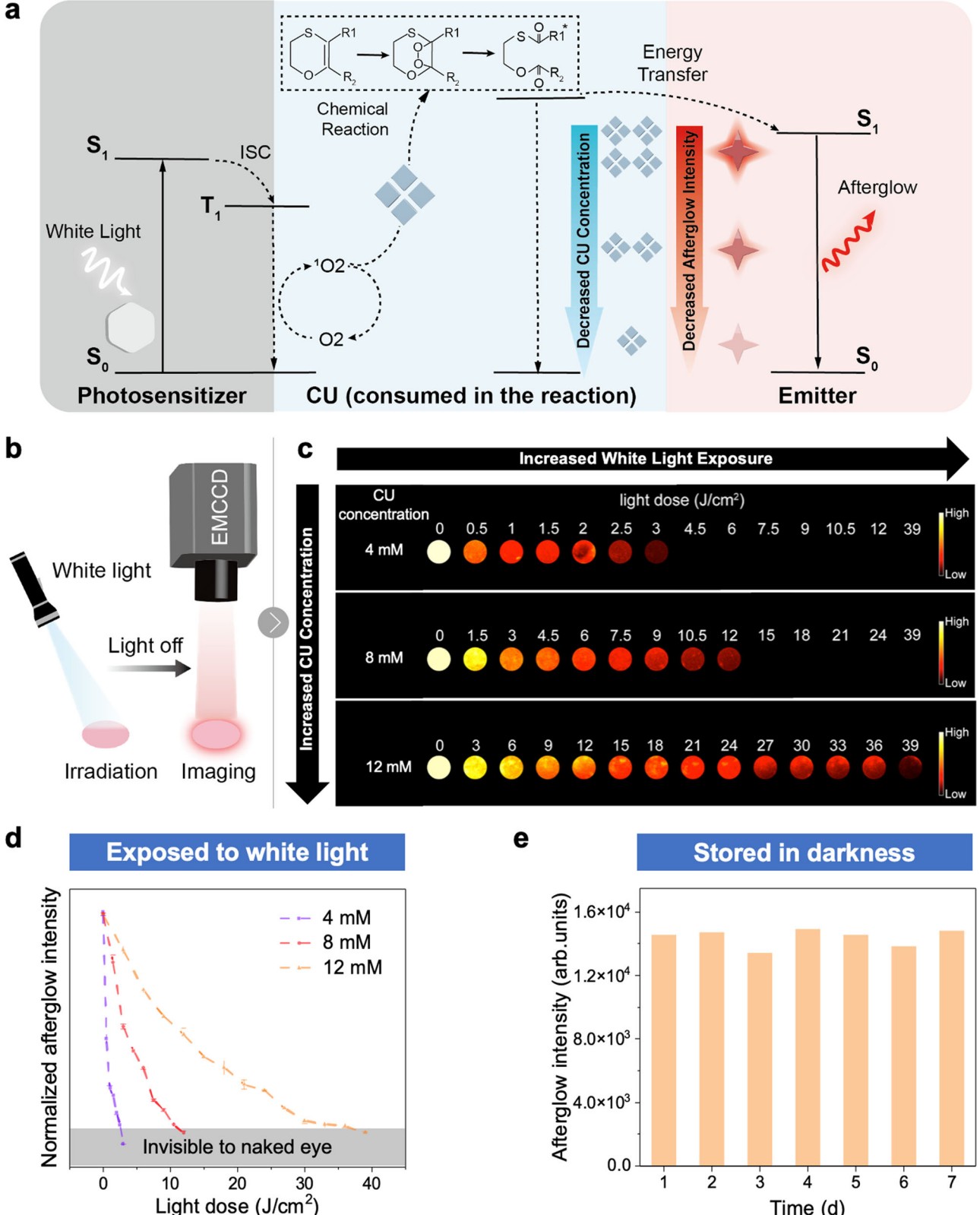

**Fig. 3 | Mechanism and features of bipolar information indication system to indicate changing information. a** Mechanism of the photochemical reaction triggered by white light to indicate changing information, CU refers to consumption units. **b** Schematic diagram of recording the afterglow intensity of bipolar PC film with EMCCD (electron-multiplying charge-coupled device). **c** Afterglow imaging of bipolar PC films with different CU concentration and degree of white light exposure. **d** Afterglow intensity of bipolar PC films with different concentrations of CU under different white light doses. **e** Afterglow intensity of bipolar PC film at a CU concentration of 12 mM stored in darkness for different time periods.

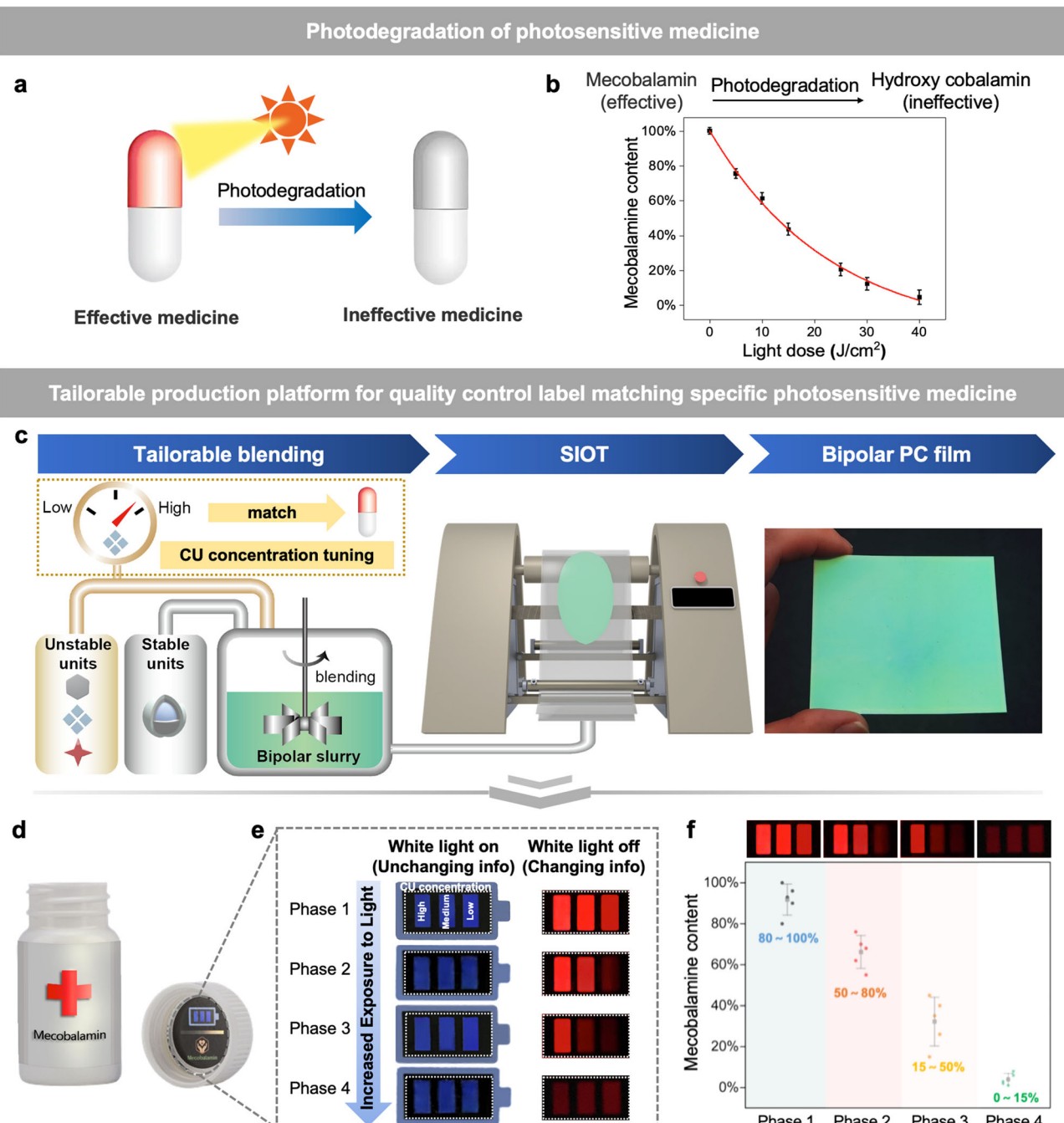

**Fig. 4 | Tailorable quality control label for photosensitive medicines based on the bipolar information indication system enabled by SIOT. a** Schematic diagram of photodegradation of photosensitive medicine (red pill refers to effective medicine while gray pill ineffective after exposed to excessive light). **b** Photodegradation curve of mecobalamin (a photosensitive medicine) under white light exposure. Error bars represent mean ± standard deviations. $n = 5$. **c** Mass production of bipolar PC films tailored to specific photosensitive medicine enabled by SIOT in the laboratory. In the left panel, low refers to CU concentration at 4 mM and high 12 mM. In the center panel, the machine can shear the disordered bipolar slurry prepared from left panel and form ordered bipolar PC films. The right panel shows the reflection color of the prepared PC film on a black background. **d** Schematic of a quality control label placed on photosensitive medicine. **e** Pictures of quality control label storing unchanging and changing information of the photosensitive medicine under different lighting conditions. (The CU concentration in the label from left to right, High: 12 mM, Medium: 8 mM, Low: 4 mM). **f** Remaining mecobalamin content in different phases of the quality control label. Error bars represent mean ± standard deviations. $n = 5$.

To indicate the changing and unchanging information of mecobalamin, the photosensitive label was placed on the suitable position of the pill box (Fig. 4d). As shown in Fig. 4e, under white light, the color was determined by the stable physical structure of a photonic crystal, which can be used to indicate the unchanging information of the medicine, like the name of the medicine. Also, the iridescent color enabled by the underlying sophisticated physical structure is hard to

be duplicated, adding an anti-counterfeiting feature to the label (Supplementary Fig. 17). When the white light is off, the afterglow intensity is decided by the unstable photochemical reaction, which can be used to indicate the changing information of the medicine, like the degree of white light exposure. To semi-quantitatively indicate the white-light exposure, we employed three gradient concentrations of CU (Fig. 4e, High→Low), as shown on the label from left to right (High:

12 mM, Medium: 8 mM, Low: 4 mM), which can reflect different degrees of white light exposure. As demonstrated by the previous experiment, different concentrations of CU correspond to different degrees of light exposure the label can take before its afterglow intensity can no longer be observed by the naked eye. Therefore, the afterglow at the grid with the lowest concentration of CU was first extinguished after white light exposure. As the white light exposure to the medicine increases, the afterglow of the grid with a medium concentration of CU disappears, followed by the highest concentration of CU. To further understand the relative content range of remaining mecobalamin regarding four different phases, we conducted multiple experiments and documented the results in Fig. 4f. In phase one, when the user of the label observes three luminous grids on the label, the content range of remaining mecobalamin was about 80 ~ 100%; in phase two, two luminous grids 50 ~ 80% remaining mecobalamin content; in phase three, one luminous grid 15 ~ 50% remaining mecobalamin content while in phase four when no luminous grid can be observed, the remaining mecobalamin would be 0 ~ 15%. Therefore, the quality control label enables patients to determine the efficacy of medicine at home, where the use of conventional assays to identify efficacy is impractical due to associated costs and time constraints. Specifically, this method offers significant time savings compared to conventional assays such as HPLC. HPLC test typically involves three steps (sample preparation, elution, and concentration calculation) and takes at least 1 h. In contrast, our method involves only one step (concentration estimation) and can be completed in ~2 min (Supplementary Fig. 18), making it a promising alternative. In summary, we prepared a bipolar information indication system capable of indicating changing and unchanging information without interfering with one another. This system ingeniously utilized white light in two separate ways: chemical and physical ones owning to the underlying unstable photochemical afterglow material and stable photonic crystal structure. Such a system features high color tunability, as its reflection color and afterglow luminous color can be independently programmed to indicate different information. We found that SIOT, a high-throughput assembling technique, could assemble stable units and unstable units into a photonic structure within 10 s. As a proof of concept, a quality control label for photosensitive medicine was designed to indicate the changing and unchanging information of the medicine. The CU concentration in the photochemical afterglow material can be arbitrarily tuned to indicate the degree of white light exposure of any specific photosensitive medicine. For example, a low concentration of CU can be used to match medicine with high photosensitivity while a high concentration of CU low photosensitivity. All in all, we designed a tailorable mass production platform for information indication labels through SIOT, capable of supporting a tailorable display of both unchanging and changing information, and is a promising alternative to instantly determine the efficacy of medicine at home where conventional assays are impractical. Future studies could focus on adjusting the molecular structure of CU to fine-tune the brightness and lifetime of the afterglow material, or developing a photonic crystal matrix that interacts with the afterglow material to output a coupled optical effect, which can further improve the volume of information indication and used for more advanced application scenarios.

## Methods

### Materials
Styrene, sodium persulfate, sodium dodecyl sulfate (SDS), ethyl acrylate (EA), ethanol were purchased from Sinopharm Chemical Reagent (China, Shanghai). Butanediol diacrylate (BDDA) and allyl methacrylate (AMA) were obtained from TCI (Shanghai). Potassium hydroxide (KOH), sodium hydroxide (NaOH), aluminum oxide, and propylene carbonate were obtained from Aladdin (China, Shanghai). Dowfax2A1 was purchased from Dow Chemicals. Before emulsion polymerization, chemicals were purified to remove stabilizers. Typically, EA was extracted with 1 M sodium hydroxide solution and treated with water to a neutral state after which the remained water was removed by sodium sulfate. Styrene was treated with alumina oxide column (basic, 200-300 mesh). Water used in this project was all deionized.

Palladium (II) octaethylporphine (PdOEP, 85%) were obtained from Sigma-Aldrich. Perylene and BODIPY were purchased from J&K Scientific Co., Ltd. (1,10-Phenanthroline)tris[4,4,4-trifluoro-1-(2-thienyl)−1,3-butanedionato]europium(III) [Eu(TTA)3phen] was purchased from Tokyo Chemical Industry Co., Ltd. CU was prepared by the methods described in the literature[27].

### Synthesis of PS@PEA core-interlayer-shell (CIS) particles
Different sized CIS particles were adapted from an elaborately-designed process according to our published paper[28]. Typically, the monomers of 0.9 g styrene (St) and 0.2 g butanediol diacrylate (BDDA) and 0.08 g sodium dodecyl sulfate (SDS) was used to prepare the polystyrene (PS) seeds by emulsion polymerization. Then, PS cores were formed through continuously injecting a monomer emulsion of 35.0 g St and 3.5 g BDDA into the reactor. After that, the core-interlayer structure was generated through continuously injecting a monomer emulsion composed of 12.5 g ethyl acrylate (EA) and 1.5 g allyl methacrylate (AMA) into the reactor. Finally, the core-interlayer-shell (CIS) structure was formed through continuously injecting a monomer emulsion of 23.4 g EA into the reactor. The final emulsion was later freeze-dried by a freeze drier to acquire dry CIS particles that can be further assembled into PC (photonic crystal) films featuring red structural color. For producing gold, green or blue PC films, simply change the SDS content in the first step to 0.09 g, 0.10 g and 0.13 g, respectively.

### Fabrication of white light-responsive solution
The white light-responsive solution was prepared by physically mixing photosensitizer, CU and emitters in propylene carbonate solution. In a typical experiment, the concentration of PdOEP-photosensitizer, CU and Eu-complex emitter is 5 μM, 4 mM and 2 mM, respectively. The solutions were preserved in sealed brown bottle before mixed with CIS particles. The afterglow intensity can be regulated by changing the concentration of CU.

### Fabrication of bipolar photonic crystal films through SIOT
The dry CIS particles were ground into powder using agate mortar and then blended with light-responsive solution, forming a homogeneous viscous dispersion (particle slurry), with CIS particles content at 55 wt%. Later, the homogeneous particle slurry was sandwiched between two glasses. After a few seconds of continuous pressing and shearing by the opposing glasses, with a rotation-shearing angular velocity of 1/3 rad/s and a rotation-shearing amplitude of 1/6 rad, a clear and uniform reflection color was observed. The corresponding rotation-shearing frequency was 0.5 Hz. Bright reflection color of photonic crystals was observed after a few seconds. For producing large-sized films, the SIOT process was adapted from previous work[10,29]. The bipolar particle slurry was prepared using the same process as described in the previous section. Subsequently, the bipolar particle slurry was placed between two PET films and tightly sandwiched together. The films were then placed flat on the SIOT machine (Fig. 4c). At room temperature, the air compressor was activated to facilitate film processing. The bipolar film underwent continuous bending for a total of 60 times until a bright and clear reflection color became visible. The resulting film had an approximate thickness of 180 μm.

### Fabrication of bipolar information indication label
The top three blue grids were prepared with CIS particles that reflect blue colors blended with europium complex, photosensitizer and

different concentrations of CU. From left to right, the concentration being 12 mM, 8 mM, 4 mM, respectively. The gold gradient rectangle was prepared with CIS particles that reflect gold color and a gradient black mask. The red cross pattern and the word "mecobalamin" were made with CIS particles that reflect red and green color, respectively.

## Characterizations

Absorption spectra were obtained from a Shimadzu UV-2600 spectrophotometer. Fluorescence spectra and decay curves of afterglow were measured on a fluorescence spectrometer (Edinburgh FS5 or FLS1000). Afterglow spectra were recorded by a Hamamatsu C14631 fiber optic spectrometer. Afterglow photographs were recorded by a lab-built imaging platform via an EMCCD (DU897, Andor).

The reflectance spectra were acquired using a Choptics EK2000-Pro spectrometer with an Ideaoptics FIB-Y-600-UV fiber and a Choptics LS2000-DH light source. A standard aluminum mirror, with a reflectivity of 100%, was used as the reference. Fluorescent intensity was recorded using an Edinburgh FS5 fluorescence spectrometer, and phosphorescent intensity was measured using a photonic multichannel analyzer PMA-12 from Hamamatsu (model C14631). The excitation wavelength used was 365 nm.

Optical pictures were taken using a Sigma 55 mm f/1.4 lens, with a viewing angle of 0°. The ultra-small angle X-ray scattering (USAXS) experiments were conducted at Beamline BL10U1 in the Shanghai Synchrotron Radiation Facility (SSRF) in China. The X-ray wavelength used was 0.124 nm. The USAXS detector employed was an Eiger 4 M, with the detector positioned 27600 mm away from the sample and a pixel size of 75 μm. The number of pixels on the x and y axes were 2070 and 2167 respectively. The exposure time was set to 20 s, and the obtained USAXS data was analyzed using FIT 2D software.

Transmission electron microscopy (TEM) images of particles after each emulsion polymerization step were captured using a JEOL JEM 1400 electron microscope. Dynamic light scattering (DLS) tests were performed at 25 °C using a Malvern Zetasizer Nano ZS90 instrument. The DLS measurements were conducted under He-Ne laser light source with a wavelength of 632.8 nm and a testing angle of 90°.

HPLC (high performance liquid chromatograph) was performed to calculate the content of mecobalamin using Waters e2695 HPLC. The chromatographic column was C18 reverse-phase column (SunFire, $4.6 \times 150$ mm, 5 μm). The mobile phase was acetonitrile containing 0.47 wt% sodium hexane sulfonate and acetonitrile containing 0.03 mol/L sodium dihydrogen phosphate (volume ratio 81:19). The flow velocity was 0.8 mL/min and the detection wavelength 266 nm.

## Reporting summary

Further information on research design is available in the Nature Portfolio Reporting Summary linked to this article.

# Data availability

All data supporting the findings of this study are available with the article, as well as the Supplementary Information file, or available from the corresponding authors upon request. Source data are provided with this paper[30]. Source data are provided with this paper.

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

## Acknowledgements

We thank the financial support by National Natural Science Foundation of China (Grant Nos. 52131308, 52373131, 51721002) (CCW); MOST (2022YFA1203002) (CCW); Key research and development project of Guangdong Province (Grant No. 2020B010190003) (CCW).

## Author contributions

C.C.W., M.X. and H.H.W. conceived the research. H.H.W. and H.T.L. synthesized the materials and conducted characterizations involving photonic crystals. J.M.Y. and Q.W.Y. synthesized the materials and conducted characterizations involving afterglow materials. J.Y.Y. and Y.B.W. conducted measurements involving medicine. C.C.W. and H.H.W. analyzed the data and wrote the paper.

## Competing interests

The authors declare no competing interests.
