## [Peer Review File · Nature Communications]

Firefly-inspired Bipolar Information Indication System Actuated by White LightREVIEWER COMMENTS

Reviewer #1 (Remarks to the Author):

Huang et al. have demonstrated a firefly lantern-inspired bipolar information indication system. The following suggestions should be incorporated before publication.

1. The manuscript should be corrected for grammatical errors. Such errors are present in the entire manuscript. These errors are making the manuscript incomprehensible.
 2. Authors need to make the introduction a little more concise.
 3. Authors should show the DLS data in ESI.
 4. In the Fig. S10 caption, Author should report the solvent used in the study.
- The manuscript represents an interesting work suitable for publication in this journal.

Reviewer #2 (Remarks to the Author):

This manuscript presents a firefly-inspired system for indicating bipolar information, employing ordered photonic crystal nanostructures through shear-induced ordering technique (SIOT). This approach leverages two bipolar information with two distinction color: a static reflection color for unchanging information and a dynamic luminous color for changing information. Notably, the authors demonstrate the application of this bipolar photonic crystal film for quality control monitoring of photosensitive medicine (Mecobalamin). Overall, this manuscript proves the hypothesis through very attractive and interesting experiments. While the research efforts are commendable, a few key issues need to be addressed. Specific comments are detailed below.

1. The basic concept is interesting, but the advantages of this method do not stand out when it is intended to be applied to practical fields, including translational medicine. In other words, the key question is what specific benefits bipolar information provides, and what specific areas the data can truly shine in. It is necessary to show additional strong application examples to convincingly present the advantages of this method.
2. As extending questions, two key concerns arise regarding the practical application of this method. Firstly, classical methods like absorption, fluorescence lifetime, and intensity analysis already effectively assess the chemical stability of the photosensitive material and the instability of the photoreaction. The proposed bipolar information system, therefore, does not offer a clear advantage in this domain. Secondly, the system's utility appears confined to in vitro settings, hindering direct comparison of in vitro drug efficacy with the complexities of the in vivo environment. It seems necessary for the authors to present opinions on what are the unique advantages of this method and how to effectively overcome these issues.
3. While the inherent structural stability of photonic crystals is superb, "temperature- and time-dependent" effects pose significant challenges for real applications. Therefore, an additional investigation into the time and temperature dependence of the reflection color change in RBG-blended CIS particles is crucial for assessing their practical viability.
4. In Figure 3, beyond detailing the CU concentrations, please elaborate on the influence of two critical parameters: the molar ratio of photosensitizer to CU and the molar ratio of CU to emitter, on both the afterglow efficiency and lifetime. The afterglow efficiency should be derived from the interaction dynamics of the energy donor and acceptor at various molar ratios. Additionally, calculating the lifetime (τ) would provide a generalized understanding of the energy decay characteristics.
5. While this manuscript demonstrates an interesting application in differentiating the drug quality of mecobalamin (effective) and hydroxycobalamin (ineffective), it is crucial to consider the existing options. That is, two straightforward, conventional assays—singlet oxygen measurement (changing information) and absorbance analysis (unchanging information)—can achieve this purpose. Therefore, to solidify the value proposition of this method, a clear comparison with these

established approaches is vital. Highlighting the advanced features of this bipolar information system relative to conventional assays would significantly strengthen the manuscript's impact.

6. In Fig. 4, incorporating the antioxidant effect's ability to impede drug degradation would further strengthen the demonstration of this method's potential.

7. Abbreviations for SIOT is redundant in the main text, and it is necessary to write abbreviations after full names the same as other abbreviations.

Reviewer #3 (Remarks to the Author):

The authors demonstrate a firefly-inspired bipolar information indication system based on colloidal photonic crystals and photochemical afterglow material. The photochemical afterglow material is incorporated into the photonic crystal matrix through shear-induced ordering technique, which can indicate two kinds of information. The consumption unit within the afterglow material can be tuned to match different degrees of light exposure. Finally, the authors demonstrate the application of this system in indicating the changing photo-degradation level of mecobalamin and anti-counterfeiting functionality. Overall, this work is solid, however, the novelty of the work is not clear. As known, the researches via the combination of photonic crystals and afterglow materials have been reported (*Adv. Mater*, 2018, 30, 1803362; *ACS Applied Materials & Interfaces*, 2021, 13, 41131.), and the two information indication system has also been achieved in the fluorescent and structural-color materials (*Nat. Commun.* 2021, 12, 699). Thus, the authors need to emphasize the characteristics and uniqueness of this work in particular. Moreover, the following comments and questions are suggested to be addressed.

1. Photonic crystals are promising materials for the control of light emission and have been widely investigated for the manipulation of the emission intensity and lifetime of light emitters. In the manuscript, is it feasible to control the luminescence intensity and FWHM of the afterglow materials by modulating the periodic lattice of the colloidal photonic crystals?
2. Colloidal photonic crystals possess the feature of angle-dependent structural colors. Will this property be affected when they are compounded with the afterglow materials? Will the structural color be affected? It is suggested to demonstrate it.
3. For anti-counterfeiting, the designability of the pattern is also important to avoid imitation. It is suggested to display some relevant functions.
4. How about the stability of this bipolar information indication system? Will it be affected by other factors such as temperature, ambient humidity, etc.?
5. The citations in the introduction part are insufficient. There is a lot of language that is not cited, such as: "Such material possesses two optical states under different lighting conditions, namely reflection color and fluorescence color. Both colors change simultaneously when exposed to UV irradiation and is correlated with the UV irradiation dose, which can be used to track changing information. However, the UV light would interfere with the two optical states of the materials, and hence not suitable for indicating unchanging information". It is suggested to add the relevant citations.

RESPONSE TO REVIEWERS' COMMENTS

Reviewer #1 (Remarks to the Author):

Huag et al. have demonstrated a firefly lantern-inspired bipolar information indication system. The following suggestions should be incorporated before publication

1. The manuscript should be corrected for grammatical errors. Such errors are present in the entire manuscript. These errors are making the manuscript incomprehensible.

Response: Thank you for pointing out the grammatical errors! We have double checked our manuscript and corrected the errors to make the manuscript more comprehensible.

2. Authors need to make the introduction a little more concise.

Response: Thank you for your suggestion. A concise introduction does indeed enhance comprehensibility. We have eliminated redundant parts from the introduction to improve readability. The revised part has been marked by red color.

3. Authors should show the DLS data in ESI.

Response: Thank you for your advice. DLS data was shown in Table S1. The polydispersity index (PDI) of the CIS particles was below 0.05, proving the the monodispersity of the CIS particles, which is a prerequisite for forming photonic crystal structure.

4. In the Fig. S10 caption, Author should report the solvent used in the study.

Response: The solvent used in the study was deuterated chloroform (CDCl_3), the related information has been added in Figure S10.

Reviewer #2 (Remarks to the Author):

This manuscript presents a firefly-inspired system for indicating bipolar information, employing ordered photonic crystal nanostructures through shear-induced ordering technique (SIOT). This approach leverages two bipolar information with two distinction color: a static reflection color for unchanging information and a dynamic

luminous color for changing information. Notably, the authors demonstrate the application of this bipolar photonic crystal film for quality control monitoring of photosensitive medicine (Mecobalamin). Overall, this manuscript proves the hypothesis through very attractive and interesting experiments. While the research efforts are commendable, a few key issues need to be addressed. Specific comments are detailed below.

1. The basic concept is interesting, but the advantages of this method do not stand out when it is intended to be applied to practical fields, including translational medicine. In other words, the key question is what specific benefits bipolar information provides, and what specific areas the data can truly shine in. It is necessary to show additional strong application example to convincingly present the advantages of this method.

Response: Thank you for your sincere comment. According to your suggestion, we further extract the essence of the bipolar information indication system and expand the strong potential application example in vivo. The major advantages of our system are summarized as following:

- (a) The two optical states are independent of each other, meaning the indicated changing and unchanging information would not interfere with one another. This feature is enabled by the physical and chemical utilization of white light. The periodic physical nanostructure of the photonic crystal film selectively reflects the white light. The afterglow material releases the white light energy stored in advance through a photochemical reaction. Changing information is indicated through changing afterglow intensity and unchanging information stable reflection color.
- (b) Instantly indicating the photodegradation level is important in clinical practice. Consider a scenario where a patient suspects his/her medicine to be ineffective at home. It is impractical for him/her to send the medicine for assay due to time and cost constraints. Therefore, a label that can instantly indicate the photodegradation level of the medicine is highly desired. A comparison of our method with traditional assays, such as HPLC (High Performance Liquid Chromatography), reveals substantial time savings. Traditional assays like HPLC typically entail three steps—sample preparation, elution, and concentration calculation—requiring at least an hour. In contrast, our method involves just one step—concentration estimation via label's afterglow intensity—and can be completed in approximately 2 minutes (Figure R1).

Figure R1. Schematic illustration of time required for indicating the photodegradation level of photosensitive medicine through conventional assay and bipolar information indication.

(c) Just as the reviewer mentioned in Question 2, our system has the potential to be expanded to an in vivo setting, which can monitor the light dosage reaching in vivo environment during phototherapy. To date, in situ evaluation of the efficacy of drugs in vivo is a challenging task, however, it presents an intriguing and innovative concept. For example, the phototherapy is commonly employed in clinical practice. However, the precise light dosage reaching in vivo environment cannot be measured accurately due to tissue absorption. Therefore, utilizing a stable indicator as an internal standard and a decaying indicator as a measurement for dosimetry can help calculate the light dosage reaching the in vivo environment, which can help to obtain the quantitative clinical data for scientific research, we will elaborate on it in Response 2.

We also revised our manuscript, specifically the last paragraph of introduction, results and discussion as well as the conclusion (Page 5, 17, 18), which has been marked with red color to emphasize the characteristics and uniqueness of this work.

2. As extending questions, two key concerns arise regarding the practical application of this method. Firstly, classical methods like absorption, fluorescence lifetime, and intensity analysis already effectively assess the chemical stability of the photosensitive material and the instability of the photoreaction. The proposed bipolar information system, therefore, does not offer a clear advantage in this domain. Secondly, the system's utility appears confined to in vitro settings, hindering direct comparison of in vitro drug efficacy with the complexities of the in vivo environment. It seems necessary for the authors to present opinions on what are the unique advantages of this method and how to effectively overcome these issues.

Response: Thank you for your professional feedback. As you mentioned, traditional spectral analysis methods such as absorption, fluorescence lifetime, and intensity analysis are commonly used to evaluate the composition of photosensitive drugs. These methods are accurate, however, they often require lengthy drug pre-treatment and specialized equipment. In contrast, our label indication method offers a faster and simpler way for drug users to evaluate active ingredients, even in a home setting where conventional assays are impractical (as illustrated in Response 1).

Currently, assessing the efficacy of drugs within an in vivo environment remains a challenge. Nonetheless, this presents an opportunity for an innovative approach. We propose a strategy that has the potential to extend the applicability of our system to in vivo conditions, albeit with certain specific issues that require resolution. For instance, phototherapy is widely used in clinical practice; however, accurately measuring the precise light dosage delivered to the in vivo environment is hindered by tissue absorption. To address this, we suggest employing a stable indicator as an internal reference and a decaying indicator to measure light dosage reaching the in vivo environment.

This approach is based on the principle that absolute light intensities are not directly comparable across varying measurement conditions. By leveraging the intensity ratio between the decaying indicator (such as afterglow materials) and the stable indicator (like photonic crystals), we can establish a metric that remains consistent even under diverse measurement conditions. This metric provides a reliable means to evaluate the efficacy of drugs within the in vivo environment. Furthermore, we can intentionally choose decaying and stable indicators with identical wavelengths to mitigate the impact of tissue-specific absorption. This selection is based on the understanding that tissues exhibit varying degrees of absorption across the spectrum of light wavelengths. By employing this approach, we can effectively negate the influence of wavelength-dependent absorption, thereby enhancing the accuracy of our dosimetry measurements, offering an advancement in our ability to measure and optimize in vivo therapeutic outcomes.

3. While the inherent structural stability of photonic crystals is superb, "temperature- and time-dependent" effects pose significant challenges for real applications. Therefore, an additional investigation into the time and temperature dependence of the reflection color change in RBG-blended CIS particles is crucial for assessing their practical viability.

Response: Thank you for pointing this out. Time and temperature dependence of the

reflection color change is crucial in real applications. We chose CIS particles to construct photonic crystals due to their exceptional stability under varying temperatures over extended periods. Unlike pigments that tend to fade over time, reflection color is based on the underlying periodic nanostructure, which is fixed and unchanging.

To prove its stability, we have conducted a thorough research. The prepared photonic crystal films were placed under 0°C, 25°C and 50°C for 1 month, their reflectance were documented below. We noticed that both reflection peak and intensity remained steady for 1 month, regardless of the temperature (Figure R2), the related information also has been added in Figure S6.

Figure R2. Reflectance of photonic crystal films stored under (a) 0°C, (b) 25°C and (c) 50°C for 1 month.

4. In Figure 3, beyond detailing the CU concentrations, please elaborate on the influence of two critical parameters: the molar ratio of photosensitizer to CU and the molar ratio of CU to emitter, on both the afterglow efficiency and lifetime. The afterglow efficiency should be derived from the interaction dynamics of the energy donor and acceptor at various molar ratios. Additionally, calculating the lifetime (τ) would provide a generalized understanding of the energy decay characteristics.

Response: Thanks for this valuable suggestions. We fixed the molar concentration of CU to 4 mM and changed the concentration of photosensitizer and emitter, respectively, to study their influence on afterglow efficiency and lifetime.

First, the afterglow intensity at different molar ratios of photosensitizer to CU was measured. With the increase of the molar ratio of photosensitizer to CU, the afterglow intensity first increased as more $^1\text{O}_2$ was produced and then decreased due to the strong absorption capacity of photosensitizer. Then, the afterglow intensity at different molar ratios of emitter to CU was measured. With the increase of the molar ratio of emitter to CU, the distance between CU and emitter was shortened, and the energy transfer efficiency was improved, so the afterglow intensity continued to increase. When the molar ratio of photosensitizer to CU and the molar ratio of emitter to CU were changed, the calculated afterglow lifetime were about 0.7 seconds with no significant difference

(Figure R3).

Given that the lifetime of both the excited-state photosensitizer and singlet oxygen is less than a few milliseconds, we propose that the photochemical reaction of the CU is the rate-determining step in the long-lived luminescence process at a constant temperature. This reaction can be considered a quasi-first-order reaction.

Figure R3. Afterglow decay curves of the bipolar information indication systems with different concentrations of (a) photosensitizer and (c) emitter. Afterglow intensity and lifetime of the systems under different (b) molar ratios of photosensitizer to CU and (d) the molar ratios of emitter to CU. The CU concentration was fixed at 4 mM.

5. While this manuscript demonstrates an interesting application in differentiating the drug quality of mecobalamin (effective) and hydroxycobalamin (ineffective), it is crucial to consider the existing options. That is, two straightforward, conventional assays—singlet oxygen measurement (changing information) and absorbance analysis (unchanging information)—can achieve this purpose. Therefore, to solidify the value proposition of this method, a clear comparison with these established approaches is vital. Highlighting the advanced features of this bipolar information system relative to conventional assays would significantly strengthen the manuscript's impact.

Response: Thank you for your valuable input, which significantly aids in effectively communicating the value proposition of this method.

For patients, determining the efficacy of a particular medication at home by sending it for assay is impractical, given the associated costs and time constraints.

Consequently, their only choice is to discard the medication if they suspect it to be ineffective. Our method presents a promising alternative, allowing patients to simply examine the label and promptly ascertain its efficacy.

A comparison of our method with traditional assays, such as HPLC (High Performance Liquid Chromatography), reveals substantial time savings. Traditional assays like HPLC typically entail three steps—sample preparation, elution, and concentration calculation—requiring at least an hour. In contrast, our method involves just one step—concentration estimation via label’s afterglow intensity—and can be completed in approximately 2 minutes (Figure R1).

Figure R1. Schematic illustration of time required for indicating the photodegradation level of photosensitive medicine through conventional assay and bipolar information indication.

6. In Fig. 4, incorporating the antioxidant effect's ability to impede drug degradation would further strengthen the demonstration of this method's potential.

Response: Thank you for bringing this to our attention! Incorporating an antioxidant into medication can effectively slow down degradation, thereby allowing photosensitive medication to withstand more light exposure before becoming ineffective. Consequently, by monitoring medication with an added antioxidant, we can increase the consumption unit (CU) in the bipolar information indication label to account for the extended light exposure. This adjustment enables the label to accurately reflect the degradation behavior of medication containing an antioxidant.

7. Abbreviations for SIOT is redundant in the main text, and it is necessary to write abbreviations after full names the same as other abbreviations.

Response: Thank you for your advice, according to your suggestion, we revised it to make the manuscript more concise.

Reviewer #3 (Remarks to the Author):

The authors demonstrate a firefly-inspired bipolar information indication system based on colloidal photonic crystals and photochemical afterglow material. The photochemical afterglow material is incorporated into the photonic crystal matrix through shear-induced ordering technique, which can indicate two kinds of information. The consumption unit within the afterglow material can be tuned to match different degrees of light exposure. Finally, the authors demonstrate the application of this system in indicating the changing photo-degradation level of mecobalamin and anti-counterfeiting functionality. Overall, this work is solid, however, the novelty of the work is not clear. As known, the researches via the combination of photonic crystals and afterglow materials have been reported (*Adv. Mater.*, 2018, 30, 1803362; *ACS Applied Materials & Interfaces*, 2021, 13, 41131.), and the two information indication system has also been achieved in the fluorescent and structural-color materials (*Nat. Commun.* 2021, 12, 699). Thus, the authors need to emphasize the characteristics and uniqueness of this work in particular. Moreover, the following comments and questions are suggested to be addressed.

Response: Thank you for your suggestion! We have revised our manuscript accordingly, specifically the last paragraph of introduction, results and discussion as well as the conclusion marked with red color to emphasize the characteristics and uniqueness of our work.

The first paper you referenced (*Adv. Mater.*, 2018, 30, 1803362) offers a comprehensive review of how photonic crystals (PC) influence emitter luminescence. In summary, the review highlights that placing emitters on the surface of PC enhances luminescence, whereas placing them inside or under the PC suppresses luminescence. We conducted experiments to validate this effect in next Response. However, our research diverges from this review as our focus is not on the interaction between PC and emitters. Instead, we aimed to design a material capable of indicating both changing and unchanging information. PC's stable structure produces unfading color, while afterglow materials can emit changing light intensities under different white light exposures, making them suitable for the bipolar information indication system. To our knowledge, this design concept is novel and has not been reported elsewhere.

The second paper you mentioned (*ACS Applied Materials & Interfaces*, 2021, 13, 41131) achieved persistent luminescence by inhibiting the motion and rotation of organic room-temperature phosphorescent (RTP) materials through polymer chains to

enhance the intersystem crossing (ISC), which is a photophysical process. In contrast, our afterglow materials achieve persistent luminescence based on a photochemical process, featuring energy storage and transfer among three components: photosensitizer, consumption unit (CU), and emitter. This method surpasses the limitations of solid matrices for persistent luminescent materials. Compared to the aforementioned paper, our materials offer the feasibility of tunable optical properties. For instance, the emission color of the system can be adjusted simply by replacing the emitter without requiring a redesign of the molecular structure. Furthermore, the intensity of the afterglow depends on the accumulated light exposure and can be used to indicate the degree of light exposure, a capability not achievable with conventional photophysical persistent luminescent materials. We have cited this paper in reference 26 and included a brief discussion in the introduction marked with red color (Page 4 Line 16-19).

The third paper you cited (Nat. Commun. 2021, 12, 699) demonstrates two optical states—reflection color and fluorescent color—both of which carry unchanging information. In contrast, our two optical states each convey changing and unchanging information based two different utilizations of white light. The changing afterglow intensity can indicate the degree of light exposure, serving as a label to indicate the photodegradation of photosensitive medicine. We have also cited this paper in reference 13 and included a brief discussion in the introduction marked with red color (Page 3 Line 17-20).

In fact, indicating the level of photodegradation is crucial in clinical practice. Patients will find it impractical to determine the efficacy of a medication at home by sending it for assay due to associated costs and time constraints. Consequently, their only choice is to discard the medication if they suspect it to be ineffective. Our method offers a promising alternative, enabling patients to simply examine the label and promptly ascertain its efficacy. A comparison of our method with traditional assays, such as HPLC (High Performance Liquid Chromatography), reveals substantial time savings. Traditional assays like HPLC typically involves three steps—sample preparation, elution, and concentration calculation—requiring at least an hour. In contrast, our method involves only one step—concentration estimation—and can be completed in approximately 2 minutes (Figure R1). The related information has been added in Figure S18.

Figure R1. Schematic illustration of time required for indicating the photodegradation level of photosensitive medicine through conventional assay and bipolar information indication.

We also revised our manuscript, specifically the last paragraph of introduction, results and discussion as well as the conclusion (Page 5, 17, 18), which has been marked with red color to emphasize the characteristics and uniqueness of this work.

1. Photonic crystals are promising materials for the control of light emission and have been widely investigated for the manipulation of the emission intensity and lifetime of light emitters. In the manuscript, is it feasible to control the luminescence intensity and FWHM of the afterglow materials by modulating the periodic lattice of the colloidal photonic crystals?

Response: Thank you for addressing this important issue. The luminescence intensity can be modulated by the periodic lattice of colloidal photonic crystals (PC) through the Purcell effect. In our study, afterglow materials were embedded in the PC matrix, and the luminescence intensity was suppressed when the emission peak matched the photonic bandgap. A detailed explanation is provided below.

According to the Purcell effect, when luminescent materials are incorporated within the photonic crystal and the photoluminescence peak matches the photonic bandgap, the luminescence intensities are suppressed (Advanced Materials 2007, 19, 577; Chemistry of Materials 2013, 25, 2309). As illustrated in Figure R4, when luminescent material emitting red light is incorporated within the PC film that reflects matching red light, luminescence intensity is suppressed because the emitted red light is Bragg reflected by the PC structure, redistributing the luminescence spectrum at different wavelengths. This reduces the fluorescence probe detection quantity at the photonic bandgap direction (Chemical Engineering Journal 2021, 426, 131259),

thereby suppressing the observed luminescence intensity.

Figure R4. Schematic illustration of luminescence suppression within the PC film.

To validate this theory, we conducted an experiment and measured the luminescence intensity of perylene embedded in a PC structure at different angles (Figure R5). Perylene has two luminescence peaks at 473 nm and 504 nm. As the PC has an angle-dependent effect, changing the angles would alter the photonic bandgap. From our experimental results, we found that for luminescence intensity of perylene at 473 nm, the intensity around 40°~50° was suppressed, and the corresponding photonic bandgap at this range was 464~487 nm. For luminescence intensity at 504 nm, the intensity around 60°~80° was suppressed, and the corresponding photonic bandgap at this range was 509~536 nm. These results demonstrate that in the case of emitters within the PC structure, when the luminescent peak matches the photonic bandgap, the luminescence intensity is suppressed. Additionally, we found that the full width at half maximum (FWHM) remained the same, around 45 nm at different angles.

Figure R5. Luminescence intensities of perylene embedded in PC structure at (a) 473 nm and (b) 504 nm under different angles.

2. Colloidal photonic crystals possess the feature of angle-dependent structural colors. Will this property be affected when they are compounded with the afterglow materials? Will the structural color be affected? It is suggested to demonstrate it.

Response: Thank you for raising this important issue. The reason we chose colloidal photonic crystals to indicate unchanging information is that the structural color depends on the stable underlying nanostructure, which remains unchanged even with the addition of afterglow materials (Figure R6). The reflection peak position of 3D photonic crystal can be calculated using Equation (1) (Angew. Chem. Int. Ed. 2014, 53, 3318-3335):

$$\lambda = 2d (n_{eff}^2 - \sin^2 \theta)^{1/2} \quad (1)$$

Here, λ represents the reflection peak wavelength of the film and d is the lattice constant and in our system is referred to (111) plane distance. d can be calculated from the CIS particle diameter (D) obtained from TEM images: $d = \sqrt{2/3} D$. θ is the angle of incident light and n_{eff} is the effective refractive index of the photonic crystals.

The addition of small afterglow molecules into the photonic crystal would not alter either d or n_{eff} . For " d ", which represents the (111) plane distance, it is determined by the submicron-sized CIS colloidal particles, and the afterglow materials with molecular sizes would not change " d " due to the substantial size difference. As for " n_{eff} ", the molar concentration of afterglow materials within the bipolar photonic crystal film is extremely low, approximately 4 mM, and has a negligible effect on n_{eff} .

Figure R6. Reflectance of photonic crystal films with and without afterglow materials.

3. For anti-counterfeiting, the designability of the pattern is also important to avoid imitation. It is suggested to display some relevant functions.

Response: Thank you for your advice! The structural color based on colloidal photonic crystals can be utilized for anti-counterfeiting purposes due to its angle-dependent feature, which is challenging to duplicate. As a proof of concept, we designed a bird pattern to demonstrate its anti-counterfeiting function. At different viewing angles,

birds of varying colors are displayed, attributed to the underlying nanostructure, which is difficult to replicate (Figure R7). The related information has been added in Figure S17.

Figure R7. Pictures of anti-counterfeiting patterns at different viewing angles.

4. How about the stability of this bipolar information indication system? Will it be affected by other factors such as temperature, ambient humidity, etc.?

Response: Thank you for your insightful inquiry. The stability of the bipolar information indication system is crucial for real-world applications. The prepared photonic crystal films were stored in a dark environment at different temperatures (10°C, 20°C, and 30°C, 40°C with a fixed humidity at 60%) and ambient humidities (30%, 60% and 80%, with a fixed temperature at 20°C) for 24 hours, after which their afterglow intensities were measured at room temperature. We observed that the impact of temperature and ambient humidity on the afterglow intensity was not substantial when the films were kept in a dark environment (Figure R8). This indicates that our bipolar indicating system is well-suited for indicating light exposure.

Figure R8. Afterglow intensities of bipolar information indication system at different (a) temperatures and (b) relative humidities.

5. The citations in the introduction part are insufficient. There is a lot of language that is not cited, such as: “Such material possesses two optical states under different lighting

conditions, namely reflection color and fluorescence color. Both colors change simultaneously when exposed to UV irradiation and is correlated with the UV irradiation dose, which can be used to track changing information. However, the UV light would interfere with the two optical states of the materials, and hence not suitable for indicating unchanging information”. It is suggested to add the relevant citations.

Response: Thank you for pointing this out, we have added the relevant citations, the work you quoted is from “Advanced Optical Materials 5, 1700014 (2017)”, which has been cited in reference 16. We have also added more relevant citations, like reference 12-15 and 26.

REVIEWERS' COMMENTS

Reviewer #2 (Remarks to the Author):

The authors have addressed my previous concerns and comments with sufficient data to support the advantage of this method, and the overall quality of revised manuscript has improved with reasonable explanations compared to the previous version. I think this manuscript deserves publication.

Reviewer #3 (Remarks to the Author):

The authors have addressed the comments from the reviewers, so it is recommend for publication in Nature Communications.